# Evaluation of the Abilities of Three Kinds of Copper-Based Nanoparticles to Control Kiwifruit Bacterial Canker

**DOI:** 10.3390/antibiotics11070891

**Published:** 2022-07-04

**Authors:** Ganggang Ren, Zhenghao Ding, Xin Pan, Guohai Wei, Peiyi Wang, Liwei Liu

**Affiliations:** 1State Key Laboratory Breeding Base of Green Pesticide and Agricultural Bioengineering, Key Laboratory of Green Pesticide and Agricultural Bioengineering, Ministry of Education, Center for R&D of Fine Chemicals, Guizhou University, Huaxi District, Guiyang 550025, China; renganggang92@163.com (G.R.); dingzhenghao97@163.com (Z.D.); 2School of Chemistry and Chemical Engineering, Guizhou University, Huaxi District, Guiyang 550025, China; pan2975216330@outlook.com (X.P.); weigh18985964437@163.com (G.W.)

**Keywords:** copper-based nanoparticles, disease suppression, kiwifruit bacterial canker, *Pseudomonas syringae pv. actinidiae* (Psa)

## Abstract

Kiwifruit bacterial canker caused by *Pseudomonas syringae pv. actinidiae* reduces kiwifruit crop yield and quality, leading to economic losses. Unfortunately, few agents for its control are available. We prepared three kinds of copper-based nanoparticles and applied them to control kiwifruit bacterial canker. The successful synthesis of Cu(OH)_2_ nanowires, Cu_3_(PO_4_)_2_ nanosheets, and Cu_4_(OH)_6_Cl_2_ nanoparticles were confirmed by transmission and scanning electron microscopy, energy dispersive spectroscopy, X-ray diffraction analysis, and X-ray photoelectron spectroscopy. The minimum bactericidal concentrations (MBCs) of the three nanoparticles were 1.56 μg/mL, which exceeded that of the commercial agent thiodiazole copper (MBC > 100 μg/mL). The imaging results indicate that the nanoparticles could interact with bacterial surfaces and kill bacteria by inducing reactive oxygen species’ accumulation and disrupting cell walls. The protective activities of Cu(OH)_2_ nanowires and Cu_3_(PO_4_)_2_ nanosheets were 59.8% and 63.2%, respectively, similar to thiodiazole copper (64.4%) and better than the Cu_4_(OH)_6_Cl_2_ nanoparticles (40.2%). The therapeutic activity of Cu_4_(OH)_6_Cl_2_ nanoparticles (67.1%) bested that of Cu(OH)_2_ nanowires (43.9%), Cu_3_(PO_4_)_2_ nanosheets (56.1%), and thiodiazole copper (53.7%). Their therapeutic and protective activities for control of kiwifruit bacterial canker differed in vivo, which was related to their sizes and morphologies. This study suggests these copper-based nanoparticles as alternatives to conventional bactericides for controlling kiwifruit diseases.

## 1. Introduction

Kiwifruit bacterial canker caused by *Pseudomonas syringae pv. actinidiae* (Psa) is a destructive global disease that can severely reduce kiwifruit crop yields; it has become the primary barrier to developing a global kiwifruit industry [1,2]. However, there are few commercially available agents that can be used for the prevention and control of kiwifruit bacterial canker [3,4]. The main agents registered for the control of kiwifruit bacterial canker in China are oxine-copper, thiodiazole copper, and Cu_4_(OH)_6_Cl_2_. Although antibiotics, such as streptomycin, kasugamycin, tetracycline, and polyoxins, can also be used to control this disease, their overuse will cause drug-resistant bacteria, environmental pollution, and nontarget toxicity [5,6,7,8,9]. Developing drugs that can be used for the prevention and control of kiwifruit bacterial canker is urgent.

Recently, many novel compounds or methods have been developed to control kiwifruit bacterial canker, including natural products [10,11,12,13,14,15,16,17], synthetic organic compounds [3,18,19,20,21,22,23], copper complexes [24,25], sulfur [26], bacteriophages [27,28,29], and other biological control agents [30,31]. These methods exhibited excellent control efficacies for kiwifruit bacterial canker. Nevertheless, it will take a long time for a novel bactericide developed in the laboratory to enter the marketplace. Copper-based pesticides, such as Cu(OH)_2_ and Cu_4_(OH)_6_Cl_2_, have a long history of use, are low-cost, control plant fungal and bacterial diseases well, and have a high market occupancy rate [32]. However, the excessive use of copper-based pesticides has many issues: they can impact food quality parameters, be phytotoxic, induce the development of copper-resistant strains, accumulate in the soil, and negatively affect soil biota [32]. Novel applied strategies and technological advancements in copper-based pesticides must be developed to improve their efficacy in disease control and reduce the risks of bacterial resistance and environmental damage.

Nanotechnology can improve the biological activity of pesticides and is associated with enhanced efficacy, reduced input, and lower ecotoxicity [33,34,35]. Typically, metal-based nanomaterials (NMs), including Ag, Zn, and Cu-based nanoparticles (NPs), are the most common nanopesticides, which have an excellent effect for controlling plant pathogens, including bacteria and fungus [36]. For example, Ag@dsDNA@GO composites at 100 ppm significantly reduced the severity of bacterial spot disease on tomato in a greenhouse [37]. Zno NPs showed good antibacterial activity against *Xanthomonas gardneri*, *B. cinera* at 5 × 10^−3^ M [38,39]. However, copper-based nanoparticles are cheaper than silver NPs and have more variety than zinc-based nanoparticles. Currently, copper-based nanoparticles (NPs) have been widely used in the control of plant fungal diseases [40,41,42,43,44]. CuO nanoparticles and Cu_3_(PO_4_)_2_·3H_2_O nanosheets significantly reduced fungal disease in watermelon (*Citrullus lanatus*) [43] and tomato (*Solanum lycopersicum*), respectively [44], and copper nanoparticles biosynthesized using *Streptomyces griseus* could effectively control red root rot disease in tea plants [45]. The novel seed treatment agent copper sulfide sepiolite composite (SP-CuS NC) nanoplatform efficiently controls seed-borne fungal pathogens [45]. In addition, copper-containing nanoparticles have been used to control plant bacterial diseases, such as *Xanthomonas campestris* [46] and *Pseudomonas syringae pv. tabaci*, on tobacco [47] and *Erwinia amylovora* in pear trees [48]. In conclusion, copper-based nanoparticles have good prospects for controlling plant diseases.

However, there have been few studies on the application of copper-based nanoparticles in controlling kiwifruit bacterial canker. In this study, Cu(OH)_2_ nanowires, Cu_4_(OH)_6_Cl_2_ nanoparticles, and Cu_3_(PO_4_)_2_·3H_2_O nanosheets are prepared, and their effectiveness at controlling *Pseudomonas syringae pv. actinidiae* (Psa) in vitro and in vivo are reported for the first time. We also explore the mechanisms of interactions between copper-based nanoparticles and Psa. This research should provide a basis for the application of the copper-based nanomaterials in the control of kiwifruit bacterial canker.

## 2. Results

### 2.1. Material Characterization of Copper-Based Nanoparticles

Transmission electron microscopy (TEM) images showed that the as-synthesized Cu(OH)_2_ nanoparticles had a wire morphology with average diameters of about 13 nm and lengths ranging in 202–823 nm (Appendix A); the synthesized Cu_3_(PO_4_)_2_·3H_2_O nanomaterials had a sheet-like morphology with dimensions ranging 93–497 nm (Appendix A); and the particle size distribution range of the synthesized Cu_4_(OH)_6_Cl_2_ nanoparticles was 14–51 nm, with a median value of 30 nm (Appendix A). The morphologies of these three kinds of nanoparticles were also confirmed by scanning electron microscopy (SEM). As shown by the high resolution TEM (HRTEM) image in Figure 1(A-2), the crystal lattice distance of about 0.241 nm corresponds to the (131) plane of the Cu(OH)_2_ nanowires, and the crystal lattice distance of about 0.241 nm corresponds to the (113) plane of the Cu_4_(OH)_6_Cl_2_ nanoparticles. Furthermore, the energy dispersive spectroscopy (EDS) results (Figure 1(A-4,B-4,C-4)) show that Cu(OH)_2_ nanowires contain Cu and O; Cu_3_(PO_4_)_2_·3H_2_O nanosheets contain Cu, O, and P; Cu_4_(OH)_6_Cl_2_ nanoparticles contain Cu, O, and Cl; and the copper element weight% is 71.48%, 41.57% and 67.85%, respectively. X-ray photoelectron spectroscopy (XPS) results confirmed the elemental composition of these nanoparticles. The X-ray diffraction (XRD) patterns of the Cu(OH)_2_ nanowires, Cu_3_(PO_4_)_2_·3H_2_O nanosheets, and Cu_4_(OH)_6_Cl_2_ nanoparticles were consistent with the crystal structures of PDF#80-0656, PDF#22-0548, and PDF#70-0821, respectively. Broad X-ray diffraction peaks confirmed that the crystal grain sizes became small. The grain sizes of Cu(OH)_2_ nanowires, Cu_3_(PO_4_)_2_·3H_2_O nanosheets, and Cu_4_(OH)_6_Cl_2_ nanoparticles were calculated to be 53.3 nm, 71.9 nm, and 21.1 nm, respectively, by the Scherrer formula, confirming that the solid powders were nanomaterials.

### 2.2. Antibacterial Activity and Antibacterial Behavior of Cooper-Based Nanoparticles against Psa In Vitro

The plate-counting method was used to determine the antibacterial activities of the copper-based nanoparticles against Psa. As shown in Figure 2A,B, thiodiazole copper showed almost no bactericidal capacity at the concentrations of 1.56 μg/mL and 0.78 μg/mL, whereas Cu(OH)_2_ nanowires, Cu_3_(PO_4_)_2_·3H_2_O nanosheets, and Cu_4_(OH)_6_Cl_2_ nanoparticles exhibited excellent antibacterial effects against Psa during an 8 h incubation; these three nanoparticles achieved inhibition rates of 88.5%, 91.7%, and 88.3%, respectively, at a concentration of 0.78 μg/mL. Copper-based nanoparticles exhibit unique antibacterial activity against Psa compared with their bulk forms. The bulk particles of Cu(OH)_2_, Cu_3_(PO_4_)_2_·3H_2_O, and Cu_4_(OH)_6_Cl_2_ were nearly inactive against Psa at a concentration of 1.56 μg/mL (Appendix A). These results suggest that the copper-based nanoparticles had better antibacterial activities than thiodiazole copper or their corresponding bulk compounds.

Copper-based nanoparticles have complex antibacterial mechanisms, which include releasing copper ions, damaging cell membranes, or causing reactive oxygen species’ accumulation. The copper content of Cu(OH)_2_ nanowires and Cu_4_(OH)_6_Cl_2_ nanoparticles were similar and higher than that of copper phosphate nanosheets. Appendix A shows that trace amounts of copper can be detected on the surface of bacteria treated with different nanoparticles; however, the EDX data cannot be accurately quantified to judge the different effect on bacteria among different kinds of copper-based nanoparticles. The Cu^2+^ release curve of copper-based nanoparticles in aqueous solvents (Appendix A), indicated that Cu_3_(PO_4_)_2_·3H_2_O nanosheets and Cu_4_(OH)_6_Cl_2_ nanoparticles both had a higher initial release rate of ions in water than that of Cu(OH)_2_ nanowires, the released Cu^2+^ of Cu_3_(PO_4_)_2_·3H_2_O nanosheets and Cu(OH)_2_ nanowires reaching a constant value after 6 h. However, the concentration of released Cu^2+^ in Cu_4_(OH)_6_Cl_2_ suspension still increased after 16 h. Although there are differences in the concentration of released Cu^2+^ in three kinds of nanoparticle solutions, the mass of released Cu^2+^ does not exceed 2% of the mass of nanoparticles after 16 h, and most nanoparticles still exist as nanoformulations in an aqueous solution. However, there was no significant difference in the antibacterial activity of the three nanoparticles in vitro, which may indicate that copper-based nanoparticles have a complex antibacterial mechanism. Figure 2C shows that the zeta potentials of Cu(OH)_2_ nanowires and Cu_4_(OH)_6_Cl_2_ nanoparticles are positive, whereas those of Cu_3_(PO_4_)_2_·3H_2_O nanosheets and Psa are negative. Obviously, the similar antibacterial activities of the three nanoparticles means that their antibacterial mechanisms may not be closely related to electrostatic attractions. Significantly, the zeta potential of Cu_4_(OH)_6_Cl_2_ nanoparticles solution was significantly decreased when bacteria were added to shake for 30 min, indicating that the colloidal solution was unstable. This electrostatic interface reaction implied the presence of a strong surface interaction between the nanocapsules and bacteria. TEM and SEM were used to further investigate the modes of antibacterial action of the copper-based nanoparticles. The cell wall of Psa was continuous and intact, and the cell was without a vacuole in the blank control group (Figure 3A,B). The three nanoparticles had good surface interactions with the bacteria, which means that the amino acid residues of cell wall peptidoglycans may coordinate with the metal ions of the nanoparticles [48]. The Cu(OH)_2_ nanowires could penetrate the bacterial cell wall, and bacterial membrane integrity was lost. The Cu_4_(OH)_6_Cl_2_ nanoparticles could adhere uniformly to the surface of bacteria and caused bacterial swelling and obvious vacuoles (Figure 3(A-4) and Appendix A). The bacterial cells were also severely damaged when Cu_3_(PO_4_)_2_·3H_2_O nanosheets was applied. SEM results confirmed that the three kinds of nanoparticles induced the shrinkage and depression of bacterial cells. Fluorescence microscopy results showed that, after nanoparticle treatment, PI fluorescent dye entered the bacteria and combined with DNA to emit red fluorescence, which indicates that the membrane permeability of the bacteria increased. In addition, the fluorescence probe DCFH–DA was used to study the accumulation of reactive oxygen species after the nanoparticle treatment of the bacteria (Figure 3C,D). The results show that all three types of nanoparticles could induce the accumulation of reactive oxygen species in bacteria (Appendix A). However, there is little Psa cells staining with PI to emit red fluorescence after treatment with thiodiazole copper for 8 h; the results reveal that thiodiazole copper could not damage the inner membrane of the bacteria. In addition, the ROS levels of the Psa cells did not increase after treatment with thiodiazole copper for 8 h. These results indicate that thiodiazole copper has weak interaction with bacteria and completely different antibacterial mechanism compared with copper-based nanoparticles.

### 2.3. Antibacterial Effects against Kiwifruit Bacterial Canker In Vivo

Figure 4 shows that the three types of copper-based nanoparticles have therapeutic and protective activities against kiwifruit bacterial canker in vivo. The protective activities of Cu(OH)_2_ nanowires and Cu_3_(PO_4_)_2_·3H_2_O nanosheets were 59.8% and 63.2%, respectively, which is similar to the protective activities of thiodiazole copper (64.4%); the protective activity of Cu_4_(OH)_6_Cl_2_ nanoparticles was 40.2%, which was lower than that of thiodiazole copper and the other two nanoparticles. However, the therapeutic activities of the three nanoparticles were the converse; specifically, the therapeutic activity of Cu_4_(OH)_6_Cl_2_ nanoparticles (67.1%) was better than that of Cu(OH)_2_ nanowires (43.9%), Cu_3_(PO_4_)_2_·3H_2_O nanosheets (56.1%), and thiodiazole copper (53.7%).

Figure 5 shows that the three kinds of nanoparticles can be uniformly dispersed on the leaf surface of kiwifruit in the form of nanoparticles, while the commercial bulk particles with nonuniform particle sizes were unevenly distributed on the leaf surface (Appendix A).

## 3. Discussion

There are many kinds of copper-based nanoparticles, including pure metallic Cu NPs, copper oxide nanoparticles (CuO NPs), copper sulfide nanoparticles (CuS NPs), and biosynthetic copper-based nanoparticles [49]. These copper NPs are promising materials in the management of infectious and communicable diseases in the field of biological medicine [50]. Although these nanoparticles have been widely used in the control of plant fungal diseases, few studies have been conducted on their use against plant bacterial diseases. In fact, copper-containing nanoparticles have good inhibitory activity against both Gram-negative and Gram-positive bacteria, which gives them great prospects for applicability in the field of plant bacterial disease prevention. However, we need to consider the constraints of the type of nanomaterials selected: the effectiveness of disease control, preparation cost, and environmental effects. Cu(OH)_2_ and Cu_4_(OH)_6_Cl_2_ are both important varieties of copper-based pesticides with high market shares with low environmental risk when applied rationally. Furthermore, as a novel copper-based nanomaterial, Cu_3_(PO_4_)_2_·3H_2_O nanosheets have been applied in the control of plant fungal diseases; moreover, phosphorus-containing nanoparticles are beneficial to plant growth [42,43]. The preparation of the three kinds of nanoparticles is relatively easy without adding surfactant or using high-temperature aging, and industrialization is readily achievable. In addition, Appendix A shows that the Cu_4_(OH)_6_Cl_2_ dispersion has the best colloidal stability and without precipitation in 1 h, while the Cu_3_(PO_4_)_2_·3H_2_O dispersion with the worst colloidal stability was completely precipitated in 1 h. Although it did not completely precipitate within 1 h, the solution of Cu(OH)_2_ nanowires became unstable. The zeta potential vs. time of the nanoparticles solutions further confirmed the above results (Appendix A). The preparation of Cu_3_(PO_4_)_2_·3H_2_O nanosheets required heating at high temperature, and the zeta potential of the water-dispersed solution had a low absolute value, indicating that the nanoparticles tended to aggregate in water. These factors make the practical application of Cu_3_(PO_4_)_2_·3H_2_O nanosheets in agriculture disadvantageous.

The minimum bactericidal concentration (MBC) of the three kinds of nanoparticles was 1.56 μg/mL, which was much lower than that of thiodiazole copper (MBC > 100 μg/mL, Appendix A). Compared with the antibiotics [51], natural compounds [10,11,12,13,14,15,16,17], and synthetic compounds [18,19,20,21,22,23] reported in the literature, copper-based nanoparticles also have advantages in bactericidal performance. The antibacterial mechanism of copper-based nanoparticles may be related to their compositions, morphologies, and sizes [52]. However, there was no significant difference in the antibacterial activities of the three nanoparticles in vitro. TEM results may have provided a reasonable explanation for this. Although these nanoparticles have different morphologies and sizes, at least one dimension of all these nanoparticles is <30 nm. During the process of interaction with bacteria, Cu(OH)_2_ nanowires and Cu_3_(PO_4_)_2_·3H_2_O nanosheets tend to agglomerate into large particles, but these nanoparticles can still make contact with bacteria at the nanometer scale. Cu_4_(OH)_6_Cl_2_ nanoparticles of uniform size can be evenly distributed around the bacteria, which may be why they have slightly higher antibacterial activity than the other nanoparticles. Hence, Cu_4_(OH)_6_Cl_2_ nanoparticles have more advantages than the other two nanoparticles for the controlling of kiwifruit bacterial canker.

Although conventional copper-based pesticides can kill Psa bacteria on the surface, it is difficult to kill bacteria in the inner vasculature of kiwifruit. Although thiodiazole copper is a systemic bactericide with both protective and therapeutic activities and has great advantages over traditional inorganic copper pesticides, it has lower bactericidal efficiency and a higher cost than inorganic copper pesticides. Importantly, it can cause pathological changes, follicular cell hypertrophy, and hyperplasia in the thyroid in mammals [49]. Thus, novel alternative agents need to be developed. In this study, the three kinds of copper-based nanoparticles that were evaluated effectively controlled kiwifruit bacterial canker. Interestingly, there was no significant difference in the antibacterial activities of the nanoparticles in vitro, but there were significant differences in vivo. Specifically, Cu(OH)_2_ nanowires had good protective activity but poor therapeutic activity, whereas this was reversed for Cu_4_(OH)_6_Cl_2_ nanoparticles. Cu_3_(PO_4_)_2_·3H_2_O nanosheets had good protective and therapeutic activities, and their antibacterial activity in vivo was similar to that of thiodiazole copper. The differences in protective and therapeutic activities may result from their differing compositions, morphologies, and nanoparticle sizes. Cu(OH)_2_ nanowires and Cu_3_(PO_4_)_2_·3H_2_O nanosheets of larger sizes more easily formed dense protective layers for preventing bacterial infection, while Cu_4_(OH)_6_Cl_2_ nanoparticles of smaller sizes more easily entered plants for conduction and showed better therapeutic activity. In addition, copper-based nanoparticles were uniformly distributed on the leaf surface in the form of nanoparticles but not as aggregates, which was beneficial for improving the application efficiency of copper-based pesticides. Moreover, copper-based nanoparticles showed excellent inhibitory activity against Psa in vitro, but did not provide much better control of kiwifruit bacterial canker caused by Psa than thiodiazole copper, which indicated that the antibacterial behaviors of nanoparticles differed in vitro and in vivo.

Agricultural antimicrobials may have a completely different antimicrobial mechanism in vitro and in vivo. Particularly, the interaction between pesticides and plants has an important effect on their control efficiency in vivo. Typically, nonsystemic pesticides only have protective activity, and systemic pesticides have an excellent control effect for phytopathogens both in vitro and in vivo. However, there are also special cases similar with copper-based nanoparticles, for example, bismerthiazol, streptomycin, and shenqinmycin have strong systematic properties in plant. The inhibitory effect of bismerthiazol on *Xanthomonas oryzae pv. oryzae* in vitro was much lower than that of the other two chemicals; however, the control effect of the bacterial rice leaf blight in vivo was greater with bismerthiazol than with the other two chemicals [53]. The significant difference in control effect of the three compounds in vivo and in vitro is the result of their completely different antimicrobial mechanisms. In addition, natural and synthetic plant immunity inducers, including sulfur and benzothiadiazole, have little effect on pathogens in vitro, but effectively reduce infection in vivo by activating the systemic-acquired resistance signal transduction pathway [26,54,55,56,57]. Nanoparticles, including copper phosphate and silica nanoparticles, usually have complex resistant mechanisms for plant disease in vivo, such as regulating the expression of plant defense genes or inducing systemic-acquired resistance to enhance disease resistance [44,58]. Therefore, the distribution, conduction behavior, molecular mechanism, and toxicity on kiwifruit of copper-based nanoparticles of different compositions, morphologies, and particle sizes will be further studied for use in kiwifruit disease management.

## 4. Materials and Methods

### 4.1. Materials and Synthesis

Copper sulfate pentahydrate and diethylene glycol were obtained from Shanghai Ti tan Scientific Co., Ltd. (Shanghai, China); ammonium dihydrogen phosphate was obtained from Shanghai Macklin Biochemical Co., Ltd. (Shanghai, China); Copper (II) chloride dihydrate was purchased from Tianjin Damao Chemical Reagent Factory (Tianjin, China); and ammonia solution was purchased from Tianjin Zhiyuan Chemical Reagent Co., Ltd. (Tianjin, China). All chemicals were of analytical grade and used without further purification. Copper hydroxide, copper oxychloride, and copper phosphate were obtained from Kondis Chemical (Hubei) Co., Ltd., (Wuhan, China).

We synthesized Cu(OH)_2_ nanowire using the coprecipitation method. Specifically, 30 mL of NH_3_·H_2_O (0.15 M, ddH_2_O) was added slowly into 20 mL of CuSO_4_ solution (0.2 M, ddH_2_O) with gentle stirring at room temperature; we stopped stirring immediately after adding the ammonia solution, and let the solution stand for 20 min. Then, 5.0 mL of 1.5 M NaOH aqueous solution was added slowly into the above solution with gentle stirring at room temperature; we stopped stirring immediately after adding NaOH aqueous solution, and let the solution stand for 20 min. The nanoparticles were separated by centrifugation (10,000 rpm/min) using a TGL-21M tabletop high-speed refrigerated centrifuge and washed with water three times. The product was freeze-dried in vacuum for 12 h.

Cu_3_(PO_4_)_2_·3H_2_O nanosheets were prepared using a polyol method described in the literature [42]. Specifically, 10 mL of CuCl_2_·2H_2_O solution (2.0 M, ddH_2_O) was added to 50 mL of diethylene glycol, and reflux at 140 °C for 1 h. Then, 10 mL of NH_4_H_2_PO_4_ solution (1.3 M, ddH_2_O) was then rapidly injected into the reaction mixture, and the reaction proceeded for 5 h. The nanoparticles were separated by centrifugation (10,000 rpm/min) using a TGL-21M tabletop high-speed refrigerated centrifuge and washed with ethanol and water three times. The product was freeze-dried in vacuum for 12 h.

Cu_4_(OH)_6_Cl_2_ nanoparticles were prepared in the following method. A total of 80 mL of NH_3_·H_2_O (0.4 M, ddH_2_O) aqueous solution was added slowly into 50 mL of CuCl_2_ (0.4 M, ddH_2_O) with vigorous stirring at room temperature; we kept stirring for 20 min after adding ammonia solution. The nanoparticles were separated by centrifugation (10,000 rpm/min) using a TGL-21M tabletop high-speed refrigerated centrifuge, and washed with water three times. The product was freeze-dried in vacuum for 12 h.

### 4.2. Nanomaterial Characterization

A drop of the samples well dispersed in ethanol was cast onto a piece of silicon wafer and air-dried, and then was coated with a thin gold film by ion beam sputtering apparatus. SEM images were taken on a FEI Nova Nano SEM 450 field emission scanning electron microscope at 10.0 kV, and the EDAX OCTANE SUPER—A Energy Dispersive Spectrometer (EDS) was used for the analysis of the elementary composition. 

The sample was prepared by dripping a suitable volume of the sample suspension on the copper grid and then air-dried. TEM and HRTEM images were taken on a FEI Talos F200C transmission electron microscope at 200 kV. 

The XRD patterns were obtained with a Bruker D8 Advance diffractometer equipped with a copper anode producing X-rays with a wavelength of 1.5418 Å. Data were collected in continuous scan mode from 10 to 80° with the sampling speed of 2°/min.

Thermo Scientific Escalab 250 Xi X-ray photoelectron spectroscopy (XPS) was used to study the surface chemical composition of nanoparticles.

### 4.3. Methodology for the Zeta Potential Measurement

(1)Prepare 200 μg/mL copper-based nanomaterial suspension in water, and then measure the zeta potential of the solutions at 0, 5, 10, 20, 30, and 60 min using laser particle size and zeta potential analyzer (DelsaNanoC particle analyzer, Beckman Coulter, CA, USA).(2)The zeta potential vs. time of the nanoparticle solutions.(3)Evaluation for the interaction between the bacteria and copper-based nanomaterial using the zeta potential analyzer.

Prepare 10 mL of 200 μg/mL copper-based nanomaterial suspension in water, and then add 100 μL of bacteria suspension (1 × 10^9^ CFU/mL) to incubate at 28 °C for 30 min with the shaker rate at 120 rpm. The zeta potential of the mix solutions was measured by laser particle size and zeta potential analyzer (DelsaNanoC particle analyzer, Beckman Coulter, CA, USA).

### 4.4. Dissolution Experiments of Copper-Based Nanoparticles

Triplicate samples of each Cu(OH)_2_, Cu_3_(PO_4_)_2_·3H_2_O and Cu_4_(OH)_6_Cl_2_ nanomaterials suspended in water were placed under constant agitation at 120 rpm using magnetic stirrers incubator shaker (IKA RCT, Staufen, Germany) held at 28 °C and aliquots were sampled at specific intervals (0, 0.5, 1, 2, 4, 6, and 16 h). Nanoparticles were removed from suspension by centrifugation at 10000 rpm, and the supernatant was then analyzed for Cu by atomic absorption spectrophotometer Spectr (240FS, Agilenf, Brea, CA, USA) [59].

### 4.5. Antibacterial Activity In Vitro

Antibacterial activity test of nanoparticles against Psa by plate counting method. Bacteria were grown overnight at 28 °C in Luria-Bertani medium (LB) (OD_595_ = 0.7). Cells were harvested by centrifugation, washed twice with phosphate phosphate-buffered saline (PBS, 10 mM, pH 7.2), and diluted to about concentrations of 1 × 10^6^ colony-forming units/mL. Copper-based nanoparticles were dispersed in the sterilized distilled water, and then sonicated for 30 min to obtain series of copper-based NPs suspension with concentrations of 17.86, 8.93, 4.46, 2.23, 1.12 µg/mL, respectively. In an antibacterial test, 300 μL of bacteria suspension and 700 μL of sample suspension were well mixed to obtain bacteria suspensions containing copper-based NPs with final concentrations of 12.5, 6.25, 3.12, 1.56, 0.78 µg/mL, respectively. Additionally, the mixture was incubated for 8 h under constant shaking. The resulting mixture was mixed well, serially diluted, and then 20 μL of each dilution was dispersed onto LB agar plates. Colonies on the plates were counted after incubation at 28 °C for 72 h.

### 4.6. The Interaction between Bacteria and the Nanoparticles

TEM analysis: The bacterial solution (final concentration 1 × 10^7^ CFU/mL) was mixed with the copper-based nanomaterial solution with the final concentration of 50 μg/mL and incubated at 28 °C for 8 h. Thereafter, 20 μL aliquots were dropped on a carbon film coated with a copper grid (200 mesh), dried with filter paper for 1 min, and stained with uranium acetate (1%) for 10 s. The excess uranium acetate was absorbed using a filter paper, and the samples were dried at room temperature, following which they were observed and photographed via TEM (Talos F200C, FEI, Hillsboro, FL, USA).

SEM analysis: The bacterial suspension was centrifuged at 6000 rpm and 4 °C for 1.5 min. Then, precipitate bacteria were fixed with glutaraldehyde, rinsed with PBS, dehydrated in graded ethanol (30%, 50%, 70%, 90%, and 100%), critical-point-dried using Leica CPD300, and gold-plated via ion sputtering (Cressington 108, Cressington, UK) successively before being observed via SEM (Nova NanoSEM 450, FEI, Hillsboro, FL, USA).

### 4.7. Membrane Permeability and ROS Accumulation Analysis of Bacteria Treated by Nanoparticles

The bacteria were treated with the copper-based nanomaterials for 8 h, and the bacterial suspension was centrifuged at 6000 rpm and 4 °C for 1.5 min. Thereafter, the precipitate was washed thrice with PBS to remove the excess drug, and the bacterial cells were resuspended in 100 μL PBS, followed by the addition of the PI solution (20 μg/mL, 10 μL) or CM–H_2_DCFDA (DCFH-DA, 10 μM, 2 μL), and incubated for 20 min at 37 °C in the dark. The stained bacteria were centrifuged and washed thrice with PBS and observed using fluorescence microscopy (Olympus-BX53, Olympus, Tokyo, Japan).

### 4.8. In Vivo Antibacterial Bioassay against Kiwifruit Bacterial Canker

The in vivo antibacterial activities of copper-based nanoparticles against kiwifruit bacterial canker were assessed using our previously reported method with slight modifications [21]. Thiodiazole copper was used as the positive control. A 2-year-old kiwifruit plant (variety Hong Yang Hongxin) was selected to perform this experiment. Three wounds with a width of ~0.1 cm and a depth to the phloem were made on the twig of each kiwifruit tree. For the protective assay, 10 μL of copper-based nanoparticles or thiodiazole copper solution at 200 μg/mL were added into the wounds; after 24 h, 10 μL of Psa (OD_595_ = 0.2) was inoculated into all wounds. For the curative assay, 10 μL of Psa suspension (OD_595_ = 0.2) was inoculated into all wounds first; after 24 h, 10 μL of copper-based nanoparticles or thiodiazole copper solution at 200 μg/mL was added. After that, all of the treatment groups were cultured in a climate chamber with the conditions of 14 h of lighting at 14 °C and 10 h of darkness at 10 °C, both at 85% RH. The results of the curative and protective activities were observed and measured at 14 days after inoculation.

The control efficiency *I* were calculated by the following equation:control efficiency *I* (%) = (C − T)/C × 100

In the equation, C and T are the average lesion lengths of the negative control and the treatment group, respectively.

### 4.9. Distribution of Copper-Based Nanoparticles on Kiwifruit Leaf Surfaces

The kiwifruit plants were sprayed with the nanoparticle solution, and the leaves were collected a day later. The fresh leaves on the plant were rinsed thrice with ddH_2_O and dried naturally. The samples of the drug spots and adjacent areas on the leaves were acquired with a hole puncher. The fresh leaves were adhered to the sample stage using a carbon-conductive adhesive without dehydration, drying, or gold spraying. The sample surfaces were observed via SEM at low vacuum conditions with a Helix detector at 3 kV.

### 4.10. Statistical Analysis

All the numerical results were calculated as mean ± standard deviation. A one-way analysis of variance with LSD multiple comparison tests (*p* < 0.05) was performed to demonstrate the significant differences using SPSS 18.0.

## 5. Conclusions

In summary, the Cu(OH)_2_ nanowires, Cu_3_(PO_4_)_2_·3H_2_O nanosheets, and Cu_4_(OH)_6_Cl_2_ nanoparticles had excellent antimicrobial activity against Psa, and all performed better than conventional copper-based bactericides in vitro. The chemical compositions, special morphologies, and particle sizes of the nanoparticles affected their antibacterial activity. Moreover, the nanoparticles could interact with the surfaces of Psa bacteria, causing severe injury to the cell membranes and inducing oxidative stress, ultimately resulting in their death. Importantly, the three nanoparticles exhibited different therapeutic and protective activities for the control of the kiwifruit bacterial canker in vivo, which may be closely related to the sizes and morphologies of their nanoparticles. This report is only the beginning of the application of nanoparticles in the control of kiwifruit bacterial canker. The distribution, conduction behavior, and toxicity of copper-based nanoparticles with different compositions, morphologies, and particle sizes will be further studied in kiwifruit plants. This study suggests that copper-based nanoparticles are potential alternatives to conventional bactericides for controlling kiwifruit bacterial canker.

## Figures and Tables

**Figure 1 antibiotics-11-00891-f001:**
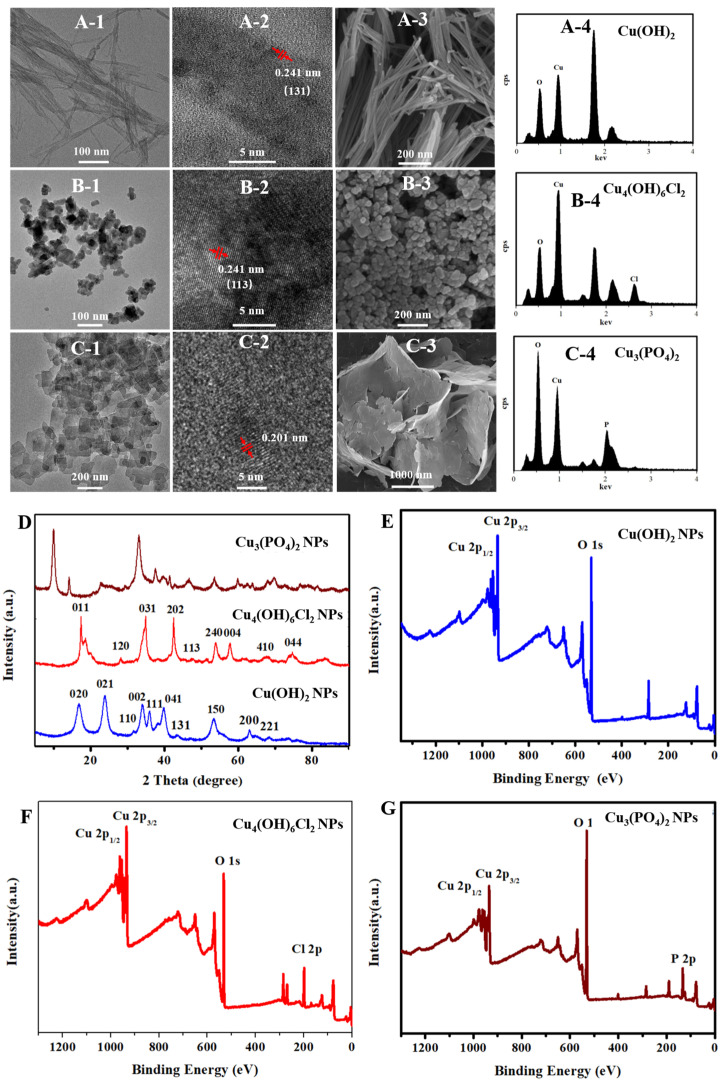
Characterization of copper-based nanomaterials. TEM (**A-1**), HR-TEM (**A-2**), SEM (**A-3**) and EDS (**A-4**) image of Cu(OH)_2_ nanowires; TEM (**B-1**), HR-TEM (**B-2**), SEM (**B-3**), and EDS (**B-4**) image of Cu_3_(PO_4_)_2_·3H_2_O nanosheets; TEM (**C-1**), HR-TEM (**C-2**), SEM (**C-3**), and EDS (**C-4**) image of copper oxychloride nanoparticles. XRD pattern of Cu(OH)_2_ nanowires, Cu_3_(PO_4_)_2_·3H_2_O nanosheets, and Cu_4_(OH)_6_Cl_2_ nanoparticles (**D**). XPS survey spectra of Cu(OH)_2_ nanowires (**E**), Cu_3_(PO_4_)_2_·3H_2_O nanosheets (**F**), and Cu_4_(OH)_6_Cl_2_ nanoparticles (**G**).

**Figure 2 antibiotics-11-00891-f002:**
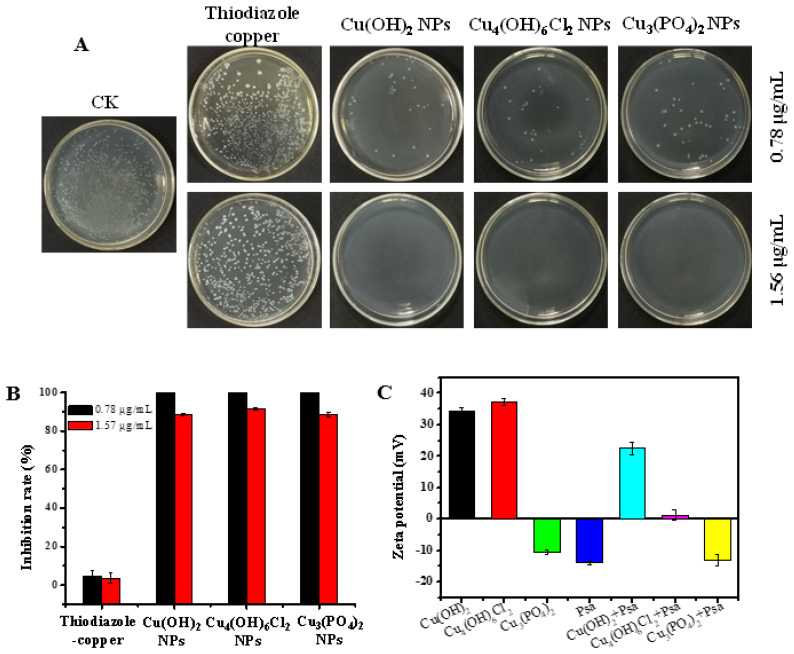
Antibacterial activity of copper-based nanomaterials. Agar plate photographs (**A**) and the inactivation efficiency (**B**) of Psa by copper-based nanomaterials was studied using a plate-counting method. Zeta potential of nanoparticles and the change of zeta potential after the interaction of nanoparticles with bacteria (**C**). CK: Negative control.

**Figure 3 antibiotics-11-00891-f003:**
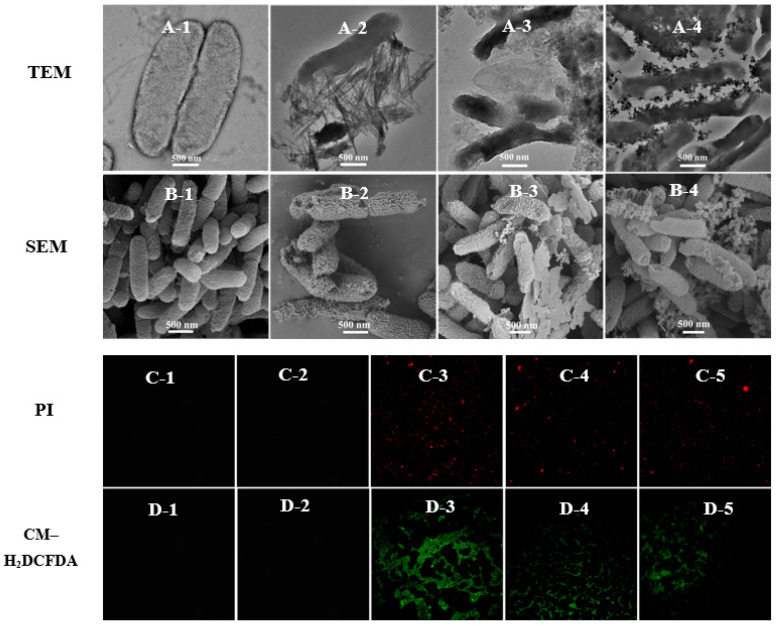
The interaction between the bacteria and nanoparticles. TEM and SEM images of Psa cells exposed to PBS (**A-1**,**B-1**), Cu(OH)_2_ nanowires (**A-2**,**B-2**), Cu_3_(PO_4_)_2_·3H_2_O nanosheets (**A-3**,**B-3**), and Cu_4_(OH)_6_Cl_2_ nanoparticles (**A-4**,**B-4**) for 8 h with a concentration of 50 μg/mL. Fluorescence images of Psa cells incubated with PI or CM–H_2_DCFDA after being exposed to PBS (**C-1**,**D-1**), thiodiazole copper (**C-2**,**D-2**), Cu(OH)_2_ nanowires (**C-3,D-3**), Cu_3_(PO_4_)_2_·3H_2_O nanosheets (**C-4,D-4**), and Cu_4_(OH)_6_Cl_2_ nanoparticles (**C-5**,**D-5**) for 8 h with concentration of 50 μg/mL.

**Figure 4 antibiotics-11-00891-f004:**
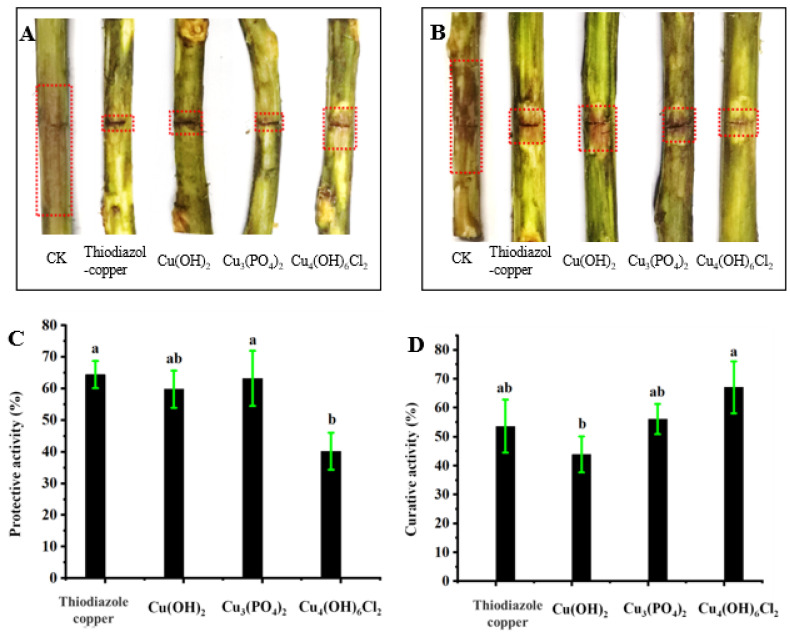
Control efficacies of copper-based nanomaterials and thiodiazole copper towards kiwifruit bacterial canker under controllable greenhouse conditions at 200 μg/mL. Protective activities (**A**,**C**); therapeutic activity (**B**,**D**). A one-way ANOVA with an LSD multiple comparison test was used to determine significance across all the treatments. Values in each panel followed by different letters are significantly different at *p* < 0.05.

**Figure 5 antibiotics-11-00891-f005:**
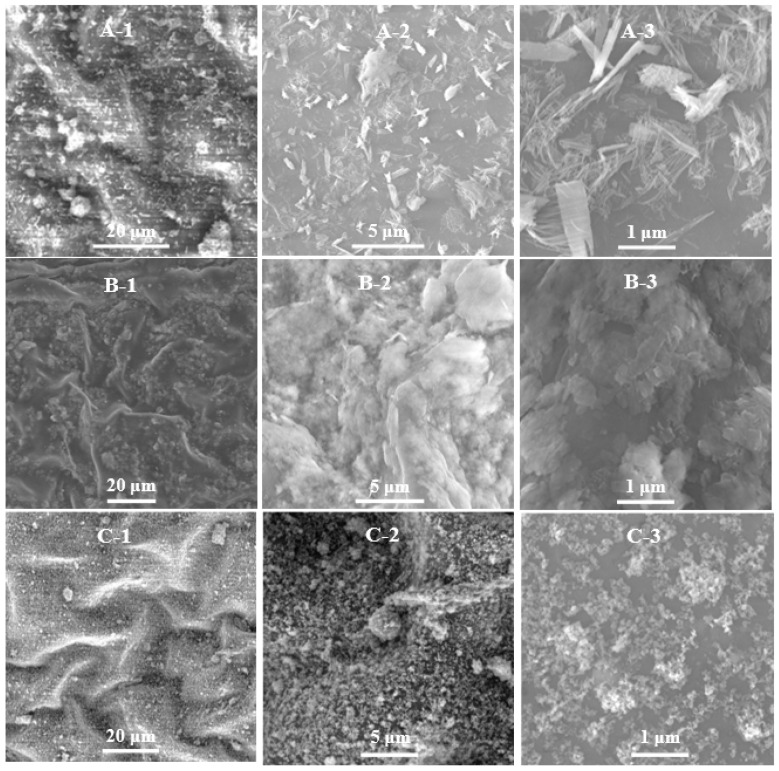
The distribution of Cu(OH)_2_ nanowires (**A-1**–**A-3**), Cu_3_(PO_4_)_2_·3H_2_O nanosheets (**B-1**–**B-3**), and Cu_4_(OH)_6_Cl_2_ nanoparticles (**C-1**–**C-3**) on the surface of kiwifruit leaves.

## Data Availability

All data generated or analyzed during this study are included in the manuscript.

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
