# Peer review of "Evaluation of the Abilities of Three Kinds of Copper-Based Nanoparticles to Control Kiwifruit Bacterial Canker"

_antibiotics, 2022, doi:10.3390/antibiotics11070891_

Round 1

Reviewer 1 Report

The presented work "Evaluation of the abilities of three kinds of copper-based nanoparticles to control kiwifruit bacterial canker" is interesting and can be considered for publication after the author(s) should consider the followings,

1- Review on the use of other nanomaterials is lacking (such as Ag or Zn nanoparticles) used as antibacterial agents with the comparison of the morphology with active dose.

2- Small editing mistakes need correction such as “Scherler formula” to be changed to “Scherrer formula” (97) and “phosphate buffffered saline” to phosphate buffered saline (299).

3- Fig. 1 D lacks crystallite plane description for Cu3(PO4)2·3H2O nanosheets.

4- Fig 2 A acronym CK is not described as well as Fig 4 C and D have acronyms a, b, ab which also should be explained.

5- Fig 3 A-B the used magnifications are not the same for all the used samples, the current presentation lacks uniformity (same magnifications next to each other) for better comparison.

6- Fig. 3 C-D additional tests are required to compare currently presented interactions between bacteria and nanoparticles with bacteria and used antibacterial agent (thiodiazole) for better understanding of the antibacterial mechanism (if low in vitro activity for thiodiazole is also confirmed in fluorescent microscopy assay).

7- Why different reagent (CuSO4 instead of CuCl2) was used for the synthesis of nanowires? Using the same reagent with different synthesis conditions would help to compare the antibacterial effect (like nanowires synthesis methodology used in 10.1134/S0036023608010063).

8- Please compare different synthesis methods in the introduction for the readers to understand the diversity of nanoparticle synthesis techniques from chemical to green (using the references, such as doi.org/10.1049/iet-nbt.2016.0238 , doi.org/10.3390/biology10080784 and doi.org/10.1049/iet-nbt.2016.0106).

9- Methodology for zeta potential measurement is missing.

10- The difference in zeta potential measurement is explained for Cu3(PO4)2·3H2O nanosheets, however, Cu4(OH)6Cl2 experience much more noticeable difference in zeta measurement potential before and after bacterial interactions. The overall zeta measurement requires further description.

11- Based on the in vitro studies, all the prepared nanoparticles have similar inhibitory effect. Nevertheless, fluorescence studies show that Cu(OH)2 nanowires generated more ROS and Cu3(PO4)2·3H2O nanosheets caused more membrane damage. It is important to provide more images from fluorescence studies to show if the effect is localized in the given area or if it is the effect of the used material.

12- The antibacterial activity in vitro and in vivo requires further explanation with existing literature.

13- The high curative activity of Cu4(OH)6Cl2 can be related with high copper content in its structure as compared with other nanomaterials (EDX). It would be useful to test the copper ions content in the treated bacteria to test that hypothesis.

14- It would be interesting for the readers to see if the combination of most efficient curative and therapeutic NP (same concentration) also shows good curative and therapeutic activity or is it decreased.

Author Response

Dear Reviewer:

Reviewer 2 Report

This manuscript “Evaluation of the abilities of three kinds of copper-based nanoparticles to control kiwifruit bacterial canker” describes copper-based nanoparticles for antibacterial applications, especially for kiwifruit bacterial canker. Overall, it needs a major revision to be published.

1. Please evaluate the colloidal stability of nanomaterials

2. Please investigate the time-dependent stability of nanoparticles or release of copper ions in the aqueous solvents, which will test whether copper ions effect the antibacterial effects.

3. Figure 3 is not indicated in the text of manuscript.

4. In Figure 4, please evaluate statistical difference with one-way ANOVA test.

5. What is the difference between protective effects and therapeutic effects? Please describe the method in more detail in the results section.

6. In vivo, what is the exact advantages of these materials over thiodiazole copper?

Author Response

Dear Reviewer:

Reviewer 3 Report

This study, which is performed using different methods, demonstrates the ability of copper-based nanoparticles to suppress the growth of Pseudomonas syringae pv. actinidiae, that is known as a cause of Kiwifruit bacterial canker. Thanks to the successful synthesis of Cu(OH)2 nanowires, Cu3(PO4)2·3H2O nanosheets, and Cu4(OH)6Cl2 nanoparticles it was possible to obtain new compounds with high antimicrobial activity that act better than conventional copper-based bactericides in vitro. The results are of interest.

There is only one drawback, which is that Fig. 3, as well as Fig. 4, were broken between the pages, which in our opinion is unfortunate, but at the same time a significant drawback, especially for Fig. 4, that prevent a clear understanding the essence of these figures.

As a whole, in spite of the manuscript is of interest, it should be published only after elimination of defect mentioned above.

Author Response

Dear Reviewer:

Round 2

Reviewer 1 Report

The manuscript "Evaluation of the abilities of three kinds of copper-based nanoparticles to control kiwifruit bacterial canker" is significantly improved after revision and can be considered for publication.

Reviewer 2 Report

The authors have successfully responded to the reviewer's concerns.